# Intensity of de novo DSA detected by Immucor Lifecodes assay and C3d fixing antibodies are not predictive of subclinical ABMR after Kidney Transplantation

Dominique Bertrand[1]*, Rangolie Kaveri[2], Charlotte Laurent[1], Philippe Gatault[3], Maïté Jauréguy[4], Cyril Garrouste[5], Johnny Sayegh[6], Nicolas Bouvier[7], Sophie Caillard[8], Luca Lanfranco[9], Antoine Thierry[10], Arnaud François[11], Françoise Hau[2], Isabelle Etienne[1], Dominique Guerrot[1], Fabienne Farce[2]

1 Nephrology Kidney Transplantation Dialysis, CHU Rouen, Rouen, France, 2 EFS Normandie, Rouen, France, 3 Nephrology Kidney Transplantation, CHU Tours, Tours, France, 4 Nephrology Kidney Transplantation, CHU Amiens, Amiens, France, 5 Nephrology Kidney Transplantation, CHU Clermont Ferrand, Clermont Ferrand, France, 6 Nephrology Kidney Transplantation, CHU Angers, Angers, France, 7 Nephrology Kidney Transplantation, CHU Caen, Caen, France, 8 Nephrology Kidney Transplantation, CHU Strasbourg, Strasbourg, France, 9 Nephrology Kidney Transplantation, CHU Brest, Brest, France, 10 Nephrology Kidney Transplantation, CHU Poitiers, Poitiers, France, 11 Pathology, CHU Rouen, Rouen, France

* dominique.bertrand@chu-rouen.fr

**Data Availability Statement:** All relevant data are within the paper.

## Abstract

De novo donor-specific antibodies (dnDSA) are associated with antibody-mediated rejection (ABMR) and allograft loss. We tested Immucor* (IM) Luminex Single-antigen beads (LSAB) assay and C3d-fixing antibodies in the setting of dnDSA and subclinical (s) ABMR. This retrospective multicentric study included 123 patients biopsied because of the presence of subclinical de novo DSA detected by One Lamda* Labscreen (MFI > 1000). In 112 patients, sera of the day of the biopsy were available and tested in a central lab with IM Lifecodes LSAB and C3d fixing antibodies assays. In 16 patients (14.3%), no DSA was detected using Immucor test. In 96 patients, at least one DSA was determined with IM. Systematic biopsies showed active sABMR in 30 patients (31.2%), chronic active sABMR in 17 patients (17.7%) and no lesions of sABMR in 49 KT recipients (51%). Intensity criteria (BCM, BCR and AD-BCR) of DSA were not statistically different between these 3 histological groups. The proportion of patients with C3d-fixing DSA was not statistically different between the 3 groups and did not offer any prognostic value regarding graft survival. Performing biopsy for dnDSA could not be guided by the intensity criteria of IM LSAB assay. C3d-fixing DSA do not offer added value.

## Introduction

Antibody mediated rejection (AMR) is a major cause of graft loss after kidney transplantation (KT) [1] and is mainly associated with preformed anti-HLA donor specific antibodies (DSA)

**Funding:** The authors received no specific funding for this work.

**Competing interests:** The authors have declared that no competing interests exist.

**Abbreviations:** eGFR, estimated glomerular filtration rate; MDRD, Modification of Diet in Renal Disease; KT, kidney transplantation; ABMR, antibody mediated rejection; sABMR, subclinical ABMR; DSA, donor specific antibodies; dnDSA, de novo DSA; iDSA, immunodominant DSA; sDSA, sum of the DSA; CNI, calcineurin inhibitors; MMF, mycophenolate mofetil; S, steroids; Mtor-I, mammalian target of rapamycin inhibitors; LSAB, Luminex single antigen beads; OL, One Lambda; IM, Immucor; IHC, immunohistochemical; IF, immunofluorescent.

(developing in the presensitized patient early posttransplantation: phenotype 1) or de novo DSA (dnDSA) (developing late posttransplantation, mostly in relation to noncompliance: phenotype 2) [2]. Even if the histological lesions are similar in the 2 phenotypes, graft survival seems to be lower in case of dn DSA [3]. AMR is a process with an early stage called "subclinical AMR" (sAMR), in which histological lesions are present in the kidney graft without clinical graft dysfunction [4]. The monitoring of dnDSA post-KT paired with a systematic biopsy in case of appearance, even in the absence of graft dysfunction, is not part of a routine clinical practice in all KT centers. In a relatively large multicentric cohort [5] (n = 123 KT recipients with subclinical dnDSA), we recently reported that a systematic biopsy performed for dnDSA in the absence of graft dysfunction leads to a diagnosis of sAMR in over 40% of cases. This screening strategy could be guided by the MFI of dnDSA. Our study shows that the MFI of dnDSA on the day of the biopsy holds a prognostic value: the higher the MFI at biopsy (iDSA, >4000; sDSA, >6300), the higher the incidence of active sAMR. In the latter study, identification of dnDSA was exclusively performed with One Lambda (OL) Labscreen Luminex Single-antigen assay from OL (LSAB)). These data confirm the results of other team with comparable strategy [6–8].

To date, studies comparing the performance and accuracy to detect DSA in the posttransplantation period of the 2 currently available SAB tests (OL Labscreen SAB and Immucor (IM) Lifescodes SAB, class I and II), are very scarce. We compared recently, we reported [9] the performance of the 2 tests for the detection of DSA and the diagnosis of AMR in the posttransplant setting. This study showed a good correlation and agreement between OL MFI and IM BCM. However, the results of IM Lifescodes LSAB cannot be interpreted with OL usual criteria and OL Labscreen LSAB seems to be better for detecting low intensity antibodies to confirm AMR diagnosis, but clearly has a lower specificity in stable situations. Nevertheless, the study population was very heterogeneous mixing patients with early and late ABMR, and active or chronic active ABMR.

Thus, to add new elements to the ongoing discussion, we performed the present study to evaluate Immucor LSAB assay in a more homogenous cohort of KT recipients presenting with subclinical dnDSA, all detected by OL LSAB. We tried to verify that IM intensity criteria hold the same prognostic value found with One Lambda MFI and evaluate the value of C3d fixing DSA.

## Material and methods

### Patients

Patients from 9 French KT centers from the Spiesser group included in a previous study [5] were retrospectively enrolled based on the following inclusion criteria:

- Kidney transplant recipients over 18 years old

- dnDSA at any time post transplantation, defined as any DSA detected with OL Labscreen LSAB after transplant that reached MFI >1000 and absent on the day of KT [5].

- Kidney graft biopsy performed between 2008 and 2016 within 6 months of DSA detection, without graft dysfunction (serum creatinine variation <20% above baseline between DSA occurrence and DSA detection and proteinuria/creatininuria ratio <0.5 g/g).

- No specific treatment for dnDSA detection begun before kidney biopsy.

Basic demographic information, date of KT, and immunosuppressive therapy were recorded from medical charts and from the ASTRE database from the Spiesser group (final agreement from the French commission of the CNIL, decision DR-2012-518, October 29,

2012). We recorded the histology of kidney biopsies, DSA information, as well as patient and graft survival. We calculated eGFR using the Modification of Diet in Renal Disease equation [10] at 1, 3, and 5 years after biopsy, when patients reached this point. We defined graft failure as eGFR <5 mL/min/1.73 m2 and/or return to chronic dialysis.

## Assessment of dnDSA with IM Lifecodes LSAB

We tested in our central lab all patients with available sera of the day of the biopsy with IM Lifecodes LSAB class 1 and class 2.

Briefly, 10 μL of each serum sample and 40 μL of HLA class I or class II LSA were mixed and incubated in the dark for 30 minutes at room temperature. After washing with the wash buffer, 50 μL of phycoerythrinconjugated goat antihuman IgG was added to the beads and incubated for 30 minutes in the dark at room temperature. The fluorescence intensities of the samples were measured by using a Luminex 200 system. Data analysis was performed by using Match IT software provided by the manufacturer (IM). The positive control beads were coated with human IgG and were designed to yield mean fluorescence intensity (MFI) values greater than 10000 when incubated with the positive or negative control sera. Conversely, the negative control beads typically yield low MFI values (lot specific) when treated with the positive or negative control sera. To identify positive bead reactions, the background MFI value was subtracted from the raw MFI value to generate the adjusted value 1 (background corrected MFI: BCM) for each individual bead. Adjusted value 1 was then divided by the MFI value from the calculated control (CalcCON) of its respective locus to generate adjusted value 2 (BCR: BCM divided by the raw MFI of the lowest ranked bead for a locus). The CalcCON for each locus was considered the raw MFI value of the lowest ranked antigen bead for that locus. Adjusted value 3 (AD-BCR: antigen density corrected BCR values) was generated by normalization of adjusted value 2 to the amount of antigen on each bead, as indicated in the lot-specific recording sheet. Positive results were assigned when 2 of 3 criteria (BCM > 1500, BCR > 5.0, or AD-BCR > 5.0) were fulfilled. We used the the most recent kits available, composed of 96 class I and 96 class II specific beads and incorporated 1 positive control and 1 negative control beads for each set. Identification of antibody specificity was carried out using a LABScan 200 Flow analyzer (Luminex Corporation, Austin, TX). The reagents used were LABScreen Single Antigen HLA Class I and Class II (OL, Canoga Park, CA) and LIFECODES LSA Class I and Class II (IM, Stamford, CT). The tests were carried out according to the manufacturers'instructions, and the analysis was performed with IM software. The same lot number for the Luminex products was used for all sera to avoid any lot to lot variability:

- LSAB Class I 3005909–3005872

- LSAB Class II 3006250–3006249

Immunodominant DSA (iDSA) was defined as the dnDSA with the highest intensity. The sum of the DSA (sDSA) was defined as the sum of the MFI of all dnDSA in the patient.

## DSA C3d binding assay

In the same serum previously tested with IM SAB IgG, the presence of DSA was also assayed in our central lab, using a commercially available kit which detects C3d-binding DSA to LSAB on a Luminex platform according to the manufacturer protocol (Lifecodes C3d assay; Immucor, Stamford USA). The C3d protocol is based on the same protocol as identification and the same lot (same antigen-coated beads) was used. The first step consists in the incubation of 1 μL of C3d-positive control beads with 40 μL of HLA class I and class II LSAB and 10μL of patient sera. After 30 minutes of incubation in the dark, 30μL of negative serum (from male

donors) is added (source of complement). After another 30 min of incubation in the dark and 3 washes with the wash buffer, 50 μL of phycoerythrinconjugated anti-human C3d was added to the beads and incubated for 30 minutes in the dark at room temperature. The fluorescence intensities of the samples were measured by using a Luminex 200 system. Positive results were assigned when 2 of 3 criteria (Background Adjusted MFI (Bg Adj) > 1500, R-Strength > 5.0, or BCR-Neg > 5.0) were fulfilled.

## Biopsy assessment

Kidney biopsy tissue was processed for light microscopy analysis, performed after fixation in Dubosq Brazil solution, 2-μm paraffin section, and coloration by Masson trichrome, hematoxylin & eosin, periodic acid-Schiff and Marinozzi silver stainings. C4d staining was performed on paraffin sections by immunohistochemical (IHC) analysis using a rabbit anti-human C4d polyclonal IgG, or by immunofluorescent (IF) staining performed on 4-μm frozen sections with monoclonal mouse anti-human C4d antibody. Peritubular C4d was considered positive if C4d staining was identified in at least 10% of peritubular capillaries (C4d2 or C4d3) by IF or in any peritubular capillaries by IHC (C4d score, >0). All transplant biopsies for dnDSA were blinded and reviewed for the study and scored according to the 2017 Banff classification [11] by a single experienced pathologist (A.F.).

## Ethical requirements

This study was in accordance with the Helsinki declaration, and was approved by the local ethics board for non-invasive health research (Comité d'Ethique pour la Recherche Non Interventionnelle CERNI N˚E2021-06, for the Centre de Protection des Personnes Nord-Ouest-I, Rouen University Hospital, Rouen, France), which waived the need for informed consent in this retrospective analysis.

## Statistical analysis

Statistics were performed using Statview version 5.0 (SAS Institute Inc., Brie Comte Robert, France). Quantitative variables were expressed as mean ± SD, whereas qualitative variables were expressed in numbers and percentages. Categorical variables were compared using the chi-square test, and the Student t test was used for continuous variables. A 2-sided P value of <0.05 was considered to be statistically significant. Patient and graft survival data were assessed by Kaplan–Meier analysis. Logrank test was used to compare survival between groups. To demonstrate a correlation between MFI and BCM for the DSA detected, we performed a Pearson correlation test, with a correlation coefficient (r): r < 0.25 indicating low correlation, 0.25 < r < 0.5 moderate correlation, 0.5 < r < 0.75 strong correlation, and r > 0.75 excellent correlation.

## Results

### Detection of dn DSA with IM Lifescodes SAB

Among the 123 patients of our previous study [5], 112 had sera tested with IM Lifescodes SAB, class I and II. Within these 112 sera, 16 (14.3%) were negative for DSA and 96 (85.7%) were positive for at least one DSA.

**Patients tested negative for DSA with IM Lifescodes SAB.** In the group of patients tested negative for DSA (n = 16) with IM test, mean MFI of iDSA with OL test was 3420 ± 2914 (1017–10957) and MFI of sDSA was 3700 ± 3335 (1017–11858). Table 1 shows the specificity of the DSA positive with OL and negative with IM; MFI, BCM, BCR and AD-BCR were also

**Table 1. Antigen specificity of the immunodominant DSA in patients with DSA positive with Labscreen SAB and negative with Lifecodes SAB.**

| Antigen specificity of the dn DSA | Nb of KTR | MFI (OL) | BCM (IM) | BCR (IM) | AD-BCR (IM) |
|---|---|---|---|---|---|
| A2 | 1 | 1600 | 99 | 1.2 | 1.19 |
| A3 | 1 | 1300 | 35 | 0.32 | 0.33 |
| B37 | 1 | 5802 | 0 | 0 | 0 |
| B8 | 1 | 2400 | 206 | 0.83 | 0.75 |
| DQ2 | 3 | 1017–3750–8882 | 312-0-0 | 1.13–0–0 | 1.3–0–0 |
| DQ5 | 4 | 1556–2000–2443–10957 | 35-292-13-82 | 0.86–3.21–2.33–1.64 | 0.45–3.02–2.07–1.79 |
| DQ6 | 1 | 5500 | 0 | 0 | 0 |
| DQ7 | 3 | 1608–1980–2000 | 0-167-202 | 0–1.96–1.92 | 0–1.85–1.80 |
| DR13 | 1 | 1924 | 0 | 0 | 0 |

KTR: kidney transplantation recipents; dn DSA: de novo donor specific antibody; iDSA: immunodominant DSA; MFI: mean fluorescence intensity; OL: One Lambda. SAB: single antigen beads; IM: Immucor; BCM: background corrected MFI; BCR: BCM divided by the raw MFI of the lowest ranked bead for a locus; AD-BCR: antigen density corrected BCR values.

reported. Missed iDSA were class I DSA in 4 cases and class II DSA in 12 cases. MFI of sDSA were between 1000 and 3000 in 10 cases, between 3 000 and 10 000 in 4 cases and above 10 000 in 2 cases, with OL test. In this group systematic biopsies showed active sABMR in 1 patient (6.25%) and chronic active sABMR in 2 patients (12.5%).

**Patients tested positive for DSA with IM Lifescodes SAB.** In the group tested positive for DSA (n = 96), iDSA were the same with the 2 vendors in 80 patients out of 96 (83.3%). Fig

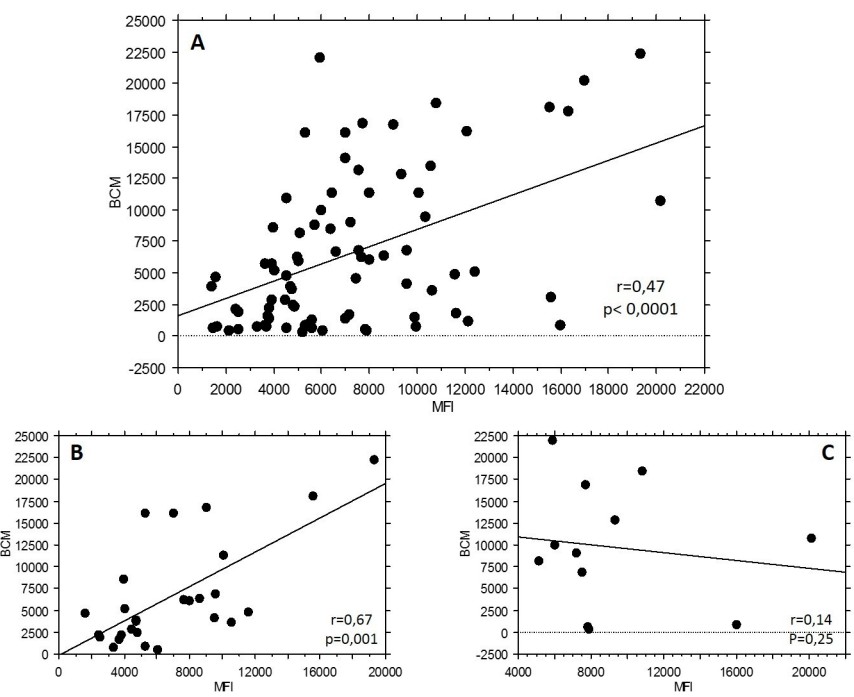

**Fig 1.** A. Correlation between BCM (Immucor Lifecodes SAB) and MFI (One Lambda Labscreen SAB) of iDSA, in the subgroup of 80 patients tested positive for the same iDSA. B. Correlation between BCM (Immucor Lifecodes SAB) and MFI (One Lambda Labscreen SAB) for anti DQ2 specificity. C. Correlation between BCM (Immucor Lifecodes SAB) and MFI (One Lambda Labscreen SAB) for anti DQ7 specificity. iDSA: immunodominant; MFI: mean fluorescence intensity; SAB: single antigen beads.

**Table 2. Baseline characteristics.**

| | n = 96 |
|---|---|
| Sex M/F n (%) | 64/32 (66.7/33.3) |
| Age (years). mean ± SD | 50.6 ± 13.1 |
| Cause of ESRD. n (%) | |
| Diabetes | 3 (3.1) |
| Glomerulonephritis | 23 (23.9) |
| Hypertension | 4 (4.2) |
| Cystic renal disease and genetic nephropathy | 31 (32.4) |
| CTIN and uropathy | 15 (15.6) |
| Other | 5 (5.2) |
| Unknown | 15 (15.6) |
| Prior solid organ transplant. n (%) | 2 (2.1) |
| Donor type. n (%) | |
| Deceased donor | 90 (93.8) |
| Living donor | 6 (6.2) |
| Class I antibodies before KT n (%) | 7 (7.3) |
| Class II antibodies before KT n (%) | 8 (8.3) |
| eGFR on the day of the biopsy ml/min/1.73 m$^2$ | 53.3 ± 18.9 |
| Proteinuria/creatininuria ratio g/g | 0.18 ± 0.15 |

ESRD: end stage renal disease; CTIN: chronic tubule-interstitial nephropathy; KT: kidney transplantation; eGFR: estimated glomerular filtration rate; M: male; F: female.

1 (panel A) shows the statistically significant but moderate correlation between BCM (IM) and MFI (OL) of iDSA, in this subgroup of 80 patients (r = 0.48). Correlation for the 2 most represented DSA, anti DQ7 (n = 27) and anti DQ2 (n = 12), was also reported in Fig 1, in panel B and C, respectively.

General characteristics of the 96 patients are presented in Table 2. Systematic biopsies showed active sABMR (group A) in 30 patients (31.2%), chronic active sABMR (group C) in 17 patients (17.7%) and no lesions of sABMR (group N) in 49 KT recipients (51%). BCM, BCR and AD-BCR of iDSA or sDSA were not statistically different between the 3 groups (Fig 2):

- for iDSA, mean BCM was 5455 ± 4685 in group A, 6428 ± 6957 in group C and 6514 ± 6261 in group N, and were not statistically different.

- for sDSA, mean BCM was 7014 ± 6080 in group A, 7255 ± 7900 in group C and 7153 ± 6654 in group N, and were not statistically different.

Fig 3 shows the proportion of the 3 histological diagnoses according to different ranges of intensity of sDSA with IM LSAB assay. The percentage of active sABMR was similar with BCM below 3000 or above 10000 (27% vs 25.7%).

## Detection of C3d-fixing DSA

In the subset of 96 patients tested positive for DSA by the 2 assays, 53 (55.2%) were positive and 43 (44.8%) were negative for C3d-fixing DSA. The proportions of patients with positive C3d-fixing DSA were not statistically different in patients with active sABMR, chronic active sABMR or without sABMR (56.7% vs 47.1% vs 57.1%, p = 0.74). In the group of 47 patients with sABMR, Banff lesions were not different between positive and negative C3d-fixing DSA and peritubular capillary C4d staining was also similar (Fig 4). In the 49 patients without

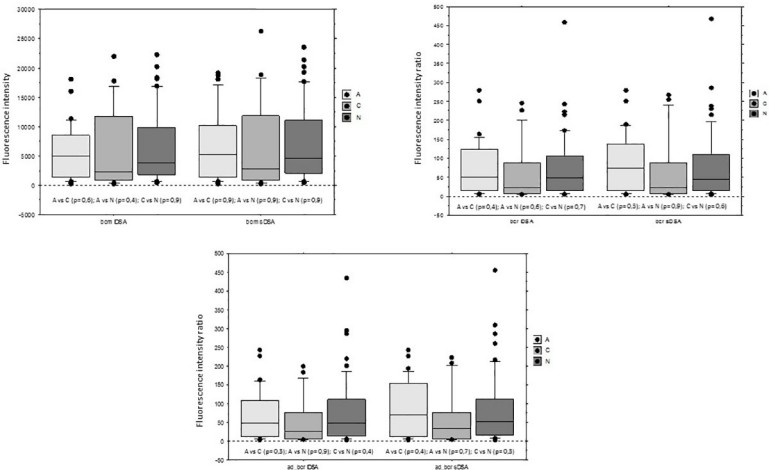

**Fig 2. BCM, BCR and AD-BCR (Lifecodes SAB) of iDSA and sDSA according to histological diagnosis on the biopsy: active sABMR (A), chronic active sABMR (C) and no sABMR (N).** sABMR: subclinical ABMR; DSA: donor specific antibodies; iDSA: immunodominant DSA; sDSA: sum of the DSA; SAB: single antigen beads. In all the box plots, the top of the box represents the 75th percentile, the bottom of the box represents the 25th percentile, and the middle line represents the 50th percentile. The whiskers represent the highest and the lowest values that are not outliers or extreme values. Black dots represent outliers.

sABMR, Banff lesions were similar but C4d staining was higher in the C3d-fxing DSA group without reaching significance (20.7% vs 5%, p = 0.08).

Death censored graft survival was not different between the 2 groups (C3d positive or negative DSA) as presented in Fig 5. eGFR was not different at 1, 3- and 5-years post biopsy according to the C3d-fixing status in the subset of patients with sABMR or without sABMR (Fig 6):

- In patients with sABMR, delta of eGFR between D0 and 5 years was -15.8 mL/min/1.73 $m^2$ in patients with C3d-binding DSA and -23.9 mL/min/1.73 $m^2$ in patients without C3d-binding DSA (p = 0.25).

- In patients without sABMR, delta of eGFR between D0 and 5 years was -3.7 mL/min/1.73 $m^2$ in patients with C3d-binding DSA and +0.6 mL/min/1.73 $m^2$ in patients without C3d-binding DSA (p = 0.37).

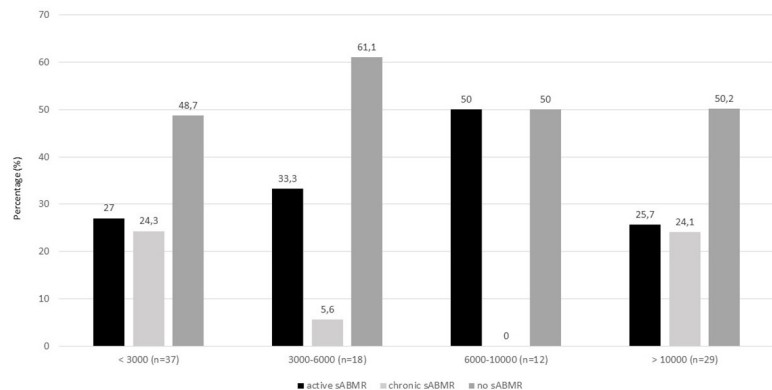

**Fig 3. Proportion of the 3 histological diagnoses according to different range of intensity (BCM) of sDSA according to Immucor Lifecodes SAB assay.** The numbers on the top of each bar refer to the percentage of each histological pattern. sDSA: sum of the DSA; SAB: single antigen beads.

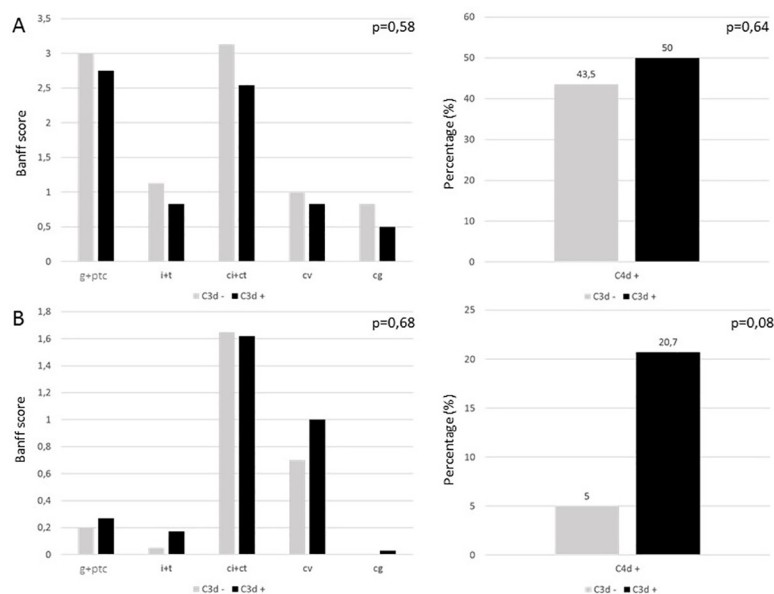

**Fig 4.** Banff lesions and peritubular capillary C4d staining according to C3d-fixing DSA status in (A) the 47 patients witch sABMR and in (B) 49 patients without sABMR. sABMR: subclinical ABMR; DSA: donor specific antibodies. For the Banff score: g: glomerulitis, i: interstitial inflammation, t: tubulitis, ptc: peritubular capillaritis, cg: allograft glomerulopathy, ci: interstitial fibrosis, ct: tubular atrophy, cv:fibrous intimal thickening. Each item is rated 0, 1, 2 or 3. For the C4d staining: each number on the top of the bar refers to the percentage of peritubular capillary C4d positive.

The intensity (BCM) of iDSA and sDSA were statistically higher in the C3d-fixing DSA group (Fig 7):

- for iDSA, mean BCM was 9020 ± 6284 in patients with C3d-fixing DSA, 2652 ± 2665 in patients without C3d-fixing DSA (p< 0.0001).

- for iDSA, mean BCM was 10087 ± 6909 in patients with C3d-fixing DSA, 3480 ± 4016 in patients without C3d-fixing DSA (p< 0.0001).

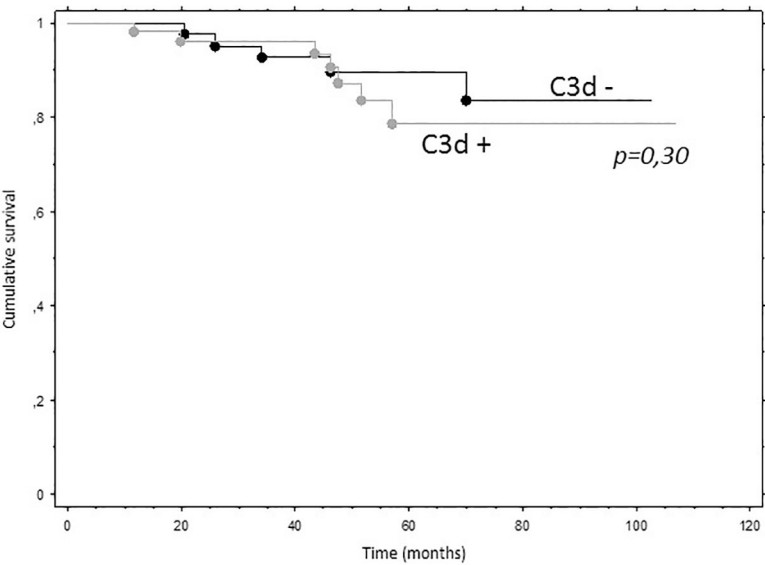

**Fig 5. Death censored graft survival according to C3d-fixing DSA status.** DSA: donor specific antibodies.

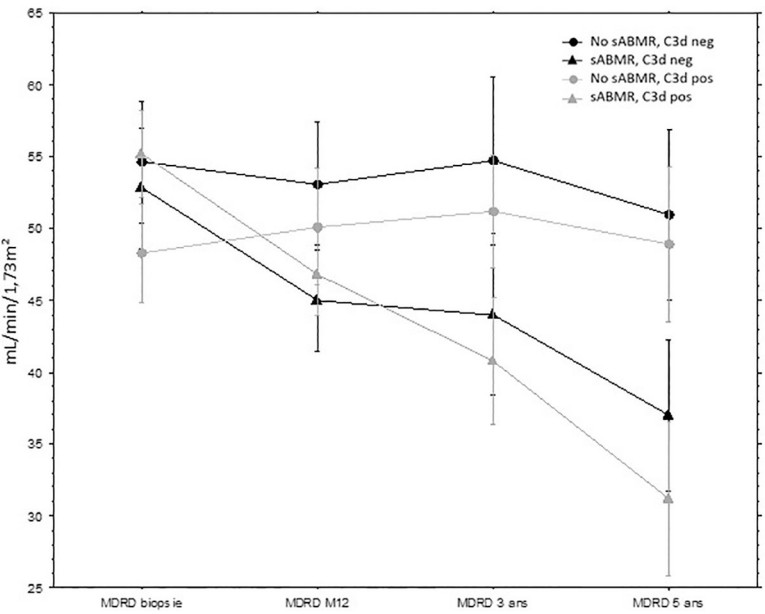

**Fig 6. eGFR at 1, 3- and 5-years post biopsy according to the C3d-fixing DSA status in the subset of patients with sABMR or without sABMR.** eGFR, estimated glomerular filtration rate; MDRD, Modification of Diet in Renal Disease; sABMR: subclinical antibody mediated rejection; DSA: donor specific antibodies.

## Discussion

To the best of our knowledge we report here the first study evaluating the LSAB from IM in a context of dnDSA detected by OL LSAB paired with systematic kidney biopsy to detect early stage of humoral rejection. Recently [5], we demonstrated, as other teams did [7, 8] that a protocol biopsy performed for a subclinical dn DSA (MFI > 1000 with OL SAB) led to the diagnosis of sABMR in up to 40% of the biospies: we showed that the diagnosis of acute sABMR was statistically associated with the MFI of iDSA and sDSA with LSAB from OL [5]. In the present

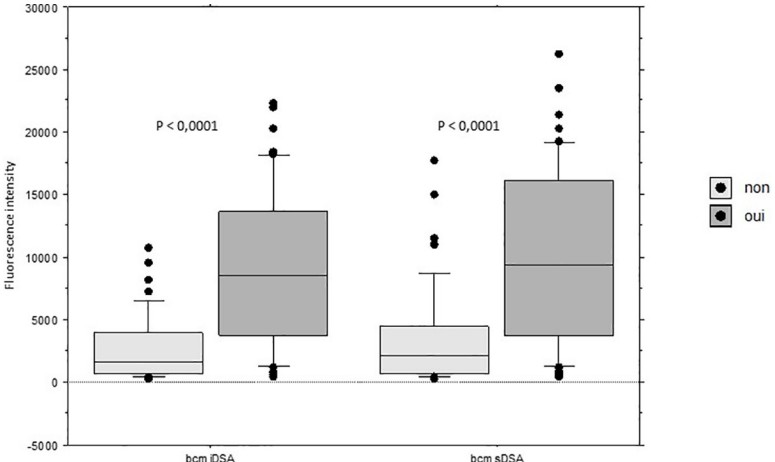

**Fig 7. Intensity (BCM) of iDSA and sDSA according to the C3d-fixing DSA status.** dnDSA: de novo DSA; iDSA: immunodominant DSA; sDSA: sum of the DSA. In all the box plots, the top of the box represents the 75th percentile, the bottom of the box represents the 25th percentile, and the middle line represents the 50th percentile. The whiskers represent the highest and the lowest values that are not outliers or extreme values. Black dots represent outliers.

study, we failed to find such association with IM test. Neither BCM, BCR nor AD-BCR were significantly different between patients with or without sABMR.

A major part of the literature on HLA antibodies, DSA detection and intensity cutoff are based on the OL test [12]. There are scarce data in the literature comparing the detection of DSA between IM and OL [13, 14]. Clerkin et al. [15] compared detection of HLA antibodies by the 2 providers in heart and lung transplant recipients. Most antibodies with moderate to high MFI titer (> 4000) were detected by both assays. Although correlation between the assays was present, significant variance was shown. Reed et al. [16] reported MFI variation to be greater with OL than IM. As We also reported that the results of Lifecodes LSAB test cannot be interpreted with OL usual criteria [9]. In the present study correlation between MFI and BCM was low, even regarding the most common specificity of DSA (r<0.5). Furthermore, in 14.3% SAB from IM did not detect any DSA. One could also argue that some class 1 DSA detected with Labscreen SAB but not with Lifecodes SAB are against denatured antigen and so, are not pathogen [17, 18]. Nevertheless, in the present study, in patients positive with Labscreen SAB but negative with Lifecodes SAB, 3 patients experienced sABMR, all in stable situation. In a study on DSA negative ABMR, Senev et al. reported a large part of DSA negative ABMR 123/208 (58.2%), possibly positive in an uncertain proportion with OL test [19]. We have to be very cautious not to interpret studies dealing with IM tests them as if they were performed with OL SAB.

Despite the fact that SAB is not a quantitative test, Schinstock et al. [8] depicted a particular risk of active sAMR associated with a sum MFI of dnDSA over 3000. In the context of pre-formed DSA, Lefaucheur et al. [20] reported that the relative risk for graft loss in patients who underwent transplantation with peak HLA–DSAs >3000 was 3.8. Wiebe et al. [21] found that dnDSA MFI sum at the time of dnDSA detection predict the risk of post-dnDSA graft loss. This suggests that performing protocol biopsy for dnDSA could be guided by the MFI of the DSA. Unfortunately, we did not observe this effect in our study with intensity criteria of Life-codes SAB: with IM, the proportion of patients with active sABMR is comparable below 3000 and above 10 000 of BCM (Fig 3). To our knowledge there is very scarce literature reporting on the predictive quantitative effect of BCM, BCR or AD-BCR on the risk of ABMR and graft survival. Recently, Senev et al. [22], reported a large cohort of 1000 KT recipients, screened for DSA with IM test. This study provided insights on the risk of ABMR and the predictive role of intensity criteria such as BCM. They validated a MFI cut-off value of 1400 units for HLA-DSA positivity proposed by the STAR working group and standardization study, and confirmed that patients with preformed DSA of background-reduced MFI <1400 had excellent allograft survival [23]. However, this should be interpreted cautiously because the thresholds depend on the vendor kit and on the instrument used, and perhaps on clinical characteristics. Therefore, it is not possible to simply generalize the thresholds found in their cohort. Furthermore, there was no cut off value regarding the risk of ABMR reported in this study.

In the last few years, there has been a large interest for complement-fixing DSA. In 69 KT recipients who fullfiled the diagnostic criteria for AMR Sicard et al. [24] reported that C3d-binding capacity of DSA at the time of AMR diagnosis allows for identification of patients at risk for allograft loss. Loupy et al. [25] reported in a larger cohort that the presence of comple-ment-binding donor-specific anti-HLA antibodies (C1q fixing complement) after transplanta-tion was associated with a risk of graft loss (hazard ratio, 4.78; 95% confidence interval [CI], 2.69 to 8.49) when adjusted for clinical, functional, histological, and immunologic factors. These antibodies were also associated with an increased rate of antibody-mediated rejection, a more severe graft injury phenotype with more extensive microvascular inflammation, and increased deposition of complement fraction C4d within graft capillaries. In 2018, a meta-analysis published by Bouquegneau et al. [26], concluded that circulating complement-

activating anti-HLA DSA had a significant deleterious impact on solid organ transplant survival and risk of rejection. Unfortunately we did not observe in the present any association between C3d-fixing DSA proportion and the diagnosis of sABMR. Furthemore, the intensity of the lesions of Banff score and C4d deposition were similar between C3d positive and C3d negative sABMR. C3d-fixing DSA were not predictive of eGFR at 5 years post biopsy nor of graft survival. In this context of subclinical dnDSA, Yamamoto et al. [6] found in a small cohort that C1q-binding DSA was a significant subclinical AMR-related factor, whereas Wiebe et a. [27] did not find any association between C1q status and AMR occurrence. Lastly, previous studies have shown a relationship between higher IgG intensity and either C3d [24] or C1q [28, 29]. We found the same correlation between BCM and C3d fixing complement. The pathogenic role of complement fixation in indolent sABMR remains to be defined.

Although our study is multicentric, with a relatively important sample, this is a retrospective study with potential bias. First of all, the fact that the Immucor tests were performed years after serum storation while the OL results were done at the time of biopsy could in a certain level of variance that might explain differences between the two systems. Secondly, no agent were added to the sera to rule out any interference of prozone / complement interference with IM LSAB. Nevertheless there is a distinct profile between the 2 tests and OL test shown clearly a more important prozone effect than IM LSAB [30]. Regarding the analysis of the sera of KT recipients with IM, a very important point is that all sera were tested by the same laboratory, experienced with the test, and overall with the same and most recent kits. It would also be interesting to test a cohort of patients with Immucor DSA + with OL test to evaluate the rate of patients with negative DSA with OL. Finally, regarding complement-fixing DSA, we did not test the sera with C1q binding DSA, which could have been interesting, to compare with C3d-biding DSA results.

In conclusion, in this relatively large multicentric study, we reported that IM Lifecodes SAB intensity criteria (BCM, BCR and AD-BCR) of DSA are not predictive of sABMR when a systematic graft biopsy is performed for dnDSA without graft dysfunction. Furthermore, a significant proportion of patients with OL positive DSA were negative with Lifecodes SAB test with clinical and histological impact. Morever, this study did not identify C3d-fixing DSA as a potential predictive factor for sABMR diagnosis or graft survival. These results suggest that identification and quantification of dnDSA with the 2 tests, OL and IM, are not similar in the setting of dnDSA monitoring post-transplant.

## Author Contributions

**Conceptualization:** Dominique Bertrand, Rangolie Kaveri, Charlotte Laurent, Philippe Gatault, Maïté Jauréguy, Cyril Garrouste, Johnny Sayegh, Nicolas Bouvier, Sophie Caillard, Luca Lanfranco, Antoine Thierry, Françoise Hau, Isabelle Etienne, Fabienne Farce.

**Data curation:** Dominique Bertrand, Rangolie Kaveri, Philippe Gatault, Maïté Jauréguy, Cyril Garrouste, Johnny Sayegh, Nicolas Bouvier, Sophie Caillard, Luca Lanfranco, Antoine Thierry, Françoise Hau, Isabelle Etienne, Fabienne Farce.

**Formal analysis:** Dominique Bertrand, Rangolie Kaveri.

**Investigation:** Dominique Bertrand, Rangolie Kaveri, Philippe Gatault, Maïté Jauréguy, Cyril Garrouste, Johnny Sayegh, Nicolas Bouvier, Sophie Caillard, Luca Lanfranco, Antoine Thierry, Arnaud François, Françoise Hau, Isabelle Etienne, Fabienne Farce.

**Methodology:** Dominique Bertrand, Rangolie Kaveri, Fabienne Farce.

**Software:** Dominique Bertrand.

**Supervision:** Dominique Bertrand.

**Validation:** Dominique Bertrand, Fabienne Farce.

**Visualization:** Dominique Bertrand.

**Writing – original draft:** Dominique Bertrand.

**Writing – review & editing:** Dominique Bertrand, Dominique Guerrot, Fabienne Farce.

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
