## [Decision Letter · Decision Letter 0]

1 Mar 2021

PONE-D-21-00625

Intensity of de novo DSA detected by Immucor Lifecodes Assay and C3d fixing antibodies are not predictive of subclinical ABMR after Kidney Transplantation.

PLOS ONE

Dear Dr. Bertrand,

Thank you for submitting your manuscript to PLOS ONE. After careful consideration, we feel that it has merit but does not fully meet PLOS ONE’s publication criteria as it currently stands. Therefore, we invite you to submit a revised version of the manuscript that addresses the points raised during the review process.

The study examined the role of de novo subclinical DSA and their potential C3-fxing antibody. Based on over 100 patients, the authors conclude that no correlation was observed between subclinical antibody mediated rejection (sAMR) and the detection of C3-fixing antibody. The reviewer # 1 requested that the authors make mor conclusive statement based on their analysis. Indeed, the statement that more study should be performed is not satisfactory as almost every study needs a continuation. The Reviewer # 2 indicated at several editorial changes which need to be addressed. 

The authors need to address points raised by the reviewers 1 and 2:

Reviewer # 1:

Bertrand and colleagues conducted a retrospective, multicenter, study to test potential correlation between the C3D-fixing antibody and IgG SAB by Imucor in a cohort of patients previously tested by the IgG SAB assay by One Lambda (MFI > 1000).

The authors confirmed previous report showing some differences between data obtained by the two vendors (Immucor and One Lambda), looking either at immunedominant DSA or sum of DSAs. They further attempted comparison between histological diagnosis on the biopsy and antibody characteristics and show no correlation

Biopsies were obtained with significant time difference from sera test (within 6 months for some samples) and the purpose for obtaining them is not clear – does not look like the same time point from transplant for all patients – the only thing in common is that they did not correlate with graft dysfunction.

Overall, this is a relatively small study cohort, utilizing patients from 9 different centers, with no systematic approach to obtaining either biopsies or serum samples. With that – it is not clear what the clinical value of their finding is. 

Reviewer # 2:

Review Comments to the Authors Manuscript title: Intensity of de novo DSA detected by Immucor Lifecodes Assay and C3d fixing antibodies are not predictive of subclinical ABMR after Kidney Transplantation. 

The rationale and objective of this study is clear. Moreover, authors successfully describe the contribution of this work to the field, while recognizing its limitations and noting outstanding questions. However, the manuscript requires much improvement to the writing quality and data presentation and interpretation. The manuscript lacks organization and focus due to run-on and awkwardly structured sentences. In addition, figures are missing crucial details required to interpret the data. 

Several results are stated without a clear connection to the supporting data. Authors should address the following points: 

Readers could benefit from a more detailed description of HLA donor specific antibodies and de novo DSA in the introduction, and how this is relevant to transplant pathology. Please explain rationale for accepting positive results for the IM Lifecodes LSAB after 2 of 3 criteria were met, instead of all requiring all criteria to be met. Please indicate negative control beads for the IM Lifecodes LSAB. 

Authors should specify concentration of beads, phycoerythrin conjugated goat antihuman IgG, anti-human C3d, instead of, or in addition to the volume used, as volume is arbitrary without knowing this information. 

Please describe methods for light microscopy and immunofluorescence staining of kidney tissue. In statistical analysis, authors should indicate a Student t-test was used. Please describe the advantages/disadvantages of the iDSA and DSA parameters. For example, why were both used, and is one superior to the other? 

Please indicate all abbreviations used in Table 1. Authors should consider reformatting the methods and results sections from bullet points to paragraphs. For reader’s ease, authors should consider labeling Figure 1 panels A-C to indicate the correlation described (as indicated in the Figure legend). For Figure 2, please define the descriptive statistics reflected in the box and whisker plot. Please label y-axes in Figures 2, 3, 4, 5, and 6. Please specific what the numbers refer to on the top of each bar in Figure 3. In addition, please explain in more detail the significance of Figure 3. Figure 4 is not clearly described and thus, difficult to interpret. To address this, please indicate the abbreviations used (g+cpt, i+t, ci+ct, cv, and cg) in the figure legend. Please define the numbers indicated on top of the bar graphs in the figure legend. 

Authors should indicate the Figure showing the data supporting the following results: “The proportions of patients with positive C3d-fixing DSA were not statistically different in patients with active sABMR, chronic active sABMR or without sABMR (56.7% vs 47.1 % vs 57.1%, p=NS).” 

Please indicate the figure showing the data supporting the following results that begin with the sentence: “The intensity (BCM) of iDSA and sDSA were statistically higher in the C3dfixing DSA group…” 

Please be consistent with reporting non-significant results as either “NS” (without a p-value) or reporting a p-value greater than 0.05. Preference is to report p-values. Authors should also consider reporting actual p-values for significant data (instead of indicating p 

The discussion should be condensed to be more concise and direct because it lacks clear focus. The discussion should also better delineate between the findings from the group’s previous work and the findings form the current study. Finally, authors should be cautious about describing data as “perfectly comparable” to previous studies. 

“Finally” is spelled incorrectly in the discussion.

We look forward to receiving your revised manuscript.

Kind regards,

Stanislaw Stepkowski

Academic Editor

PLOS ONE

Journal Requirements:

2. In the ethics statement in the manuscript and in the online submission form, please provide additional information about the patient records/samples used in your retrospective study, including: a) whether all data were fully anonymized before you accessed them; b) the date range (month and year) during which patients' medical records/samples were accessed; c) the date range (month and year) during which patients whose medical records/samples were selected for this study sought treatment; and d) the source of the medical records/samples analyzed in this work (e.g. hospital, institution or medical center name).

3.We note that your study involved tissue/organ transplantation. Please provide the following information regarding tissue/organ donors for transplantation cases analyzed in your study.

1. Please provide the source(s) of the transplanted tissue/organs used in the study, including the institution name and a non-identifying description of the donor(s).

2. Please state in your response letter and ethics statement whether the transplant cases for this study involved any vulnerable populations; for example, tissue/organs from prisoners, subjects with reduced mental capacity due to illness or age, or minors.

- If a vulnerable population was used, please describe the population, justify the decision to use tissue/organ donations from this group, and clearly describe what measures were taken in the informed consent procedure to assure protection of the vulnerable group and avoid coercion.

- If a vulnerable population was not used, please state in your ethics statement, “None of the transplant donors was from a vulnerable population and all donors or next of kin provided written informed consent that was freely given.”

3. In the Methods, please provide detailed information about the procedure by which informed consent was obtained from organ/tissue donors or their next of kin. In addition, please provide a blank example of the form used to obtain consent from donors, and an English translation if the original is in a different language.

4. Please indicate whether the donors were previously registered as organ donors. If tissues/organs were obtained from deceased donors or cadavers, please provide details as to the donors’ cause(s) of death.

5. Please provide the participant recruitment dates and the period during which transplant procedures were done (as month and year).

6. Please discuss whether medical costs were covered or other cash payments were provided to the family of the donor. If so, please specify the value of this support (in local currency and equivalent to U.S. dollars).

Thank you for your attention to these requests.

4.We note that you have indicated that data from this study are available upon request. PLOS only allows data to be available upon request if there are legal or ethical restrictions on sharing data publicly. For information on unacceptable data access restrictions, please see http://journals.plos.org/plosone/s/data-availability#loc-unacceptable-data-access-restrictions.

5.Thank you for stating the following financial disclosure:

 "NO

6.Thank you for submitting the above manuscript to PLOS ONE. During our internal evaluation of the manuscript, we found some minor occurrences of overlapping text with the following previous publication(s), some of which you are an author, which needs to be addressed:

- https://journals.lww.com/transplantjournal/Abstract/2021/03000/Complex_Linkage_Disequilibrium_Effects_in_HLA_DPB1.27.aspx?context=FeaturedArticles&collectionId=23

- https://onlinelibrary.wiley.com/doi/full/10.1111/ajt.15414

- https://www.nejm.org/doi/10.1056/NEJMoa1302506

- https://journals.lww.com/transplantjournal/Fulltext/2019/03000/Comparison_of_Two_Luminex_Single_antigen_Bead_Flow.28.aspx

We would like to make you aware that copying extracts from previous publications word-for-word, especially outside the methods section, is unacceptable. In addition, the reproduction of text from published reports has implications for the copyright that may apply to the publications.

Please revise the manuscript to quote or rephrase the duplicated text and cite your sources for text outside the methods section. Please note that further consideration is dependent on the submission of a manuscript that addresses these concerns about the overlap in text with published work.

Additional Editor Comments:

The study examined the role of de novo subclinical DSA and their potential C3-fxing antibody. Based on over 100 patients, the authors conclude that no correlation was observed between subclinical antibody mediated rejection (sAMR) and the detection of C3-fixing antibody. The reviewer # 1 requested that the authors make mor conclusive statement based on their analysis. Indeed, the statement that more study should be performed is not satisfactory as almost every study needs a continuation. The Reviewer # 2 indicated at several editorial changes which need to be addressed.

The authors need to address points raised by the reviewers 1 and 2:

Reviewer # 1:

Bertrand and colleagues conducted a retrospective, multicenter, study to test potential correlation between the C3D-fixing antibody and IgG SAB by Imucor in a cohort of patients previously tested by the IgG SAB assay by One Lambda (MFI > 1000).

The authors confirmed previous report showing some differences between data obtained by the two vendors (Immucor and One Lambda), looking either at immunedominant DSA or sum of DSAs. They further attempted comparison between histological diagnosis on the biopsy and antibody characteristics and show no correlation

Biopsies were obtained with significant time difference from sera test (within 6 months for some samples) and the purpose for obtaining them is not clear – does not look like the same time point from transplant for all patients – the only thing in common is that they did not correlate with graft dysfunction.

Overall, this is a relatively small study cohort, utilizing patients from 9 different centers, with no systematic approach to obtaining either biopsies or serum samples. With that – it is not clear what the clinical value of their finding is.

Reviewer # 2:

Review Comments to the Authors Manuscript title: Intensity of de novo DSA detected by Immucor Lifecodes Assay and C3d fixing antibodies are not predictive of subclinical ABMR after Kidney Transplantation.

The rationale and objective of this study is clear. Moreover, authors successfully describe the contribution of this work to the field, while recognizing its limitations and noting outstanding questions. However, the manuscript requires much improvement to the writing quality and data presentation and interpretation. The manuscript lacks organization and focus due to run-on and awkwardly structured sentences. In addition, figures are missing crucial details required to interpret the data.

Several results are stated without a clear connection to the supporting data. Authors should address the following points:

Readers could benefit from a more detailed description of HLA donor specific antibodies and de novo DSA in the introduction, and how this is relevant to transplant pathology. Please explain rationale for accepting positive results for the IM Lifecodes LSAB after 2 of 3 criteria were met, instead of all requiring all criteria to be met. Please indicate negative control beads for the IM Lifecodes LSAB.

Authors should specify concentration of beads, phycoerythrin conjugated goat antihuman IgG, anti-human C3d, instead of, or in addition to the volume used, as volume is arbitrary without knowing this information.

Please describe methods for light microscopy and immunofluorescence staining of kidney tissue. In statistical analysis, authors should indicate a Student t-test was used. Please describe the advantages/disadvantages of the iDSA and DSA parameters. For example, why were both used, and is one superior to the other?

Please indicate all abbreviations used in Table 1. Authors should consider reformatting the methods and results sections from bullet points to paragraphs. For reader’s ease, authors should consider labeling Figure 1 panels A-C to indicate the correlation described (as indicated in the Figure legend). For Figure 2, please define the descriptive statistics reflected in the box and whisker plot. Please label y-axes in Figures 2, 3, 4, 5, and 6. Please specific what the numbers refer to on the top of each bar in Figure 3. In addition, please explain in more detail the significance of Figure 3. Figure 4 is not clearly described and thus, difficult to interpret. To address this, please indicate the abbreviations used (g+cpt, i+t, ci+ct, cv, and cg) in the figure legend. Please define the numbers indicated on top of the bar graphs in the figure legend.

Authors should indicate the Figure showing the data supporting the following results: “The proportions of patients with positive C3d-fixing DSA were not statistically different in patients with active sABMR, chronic active sABMR or without sABMR (56.7% vs 47.1 % vs 57.1%, p=NS).”

Please indicate the figure showing the data supporting the following results that begin with the sentence: “The intensity (BCM) of iDSA and sDSA were statistically higher in the C3dfixing DSA group…”

Please be consistent with reporting non-significant results as either “NS” (without a p-value) or reporting a p-value greater than 0.05. Preference is to report p-values. Authors should also consider reporting actual p-values for significant data (instead of indicating p

The discussion should be condensed to be more concise and direct because it lacks clear focus. The discussion should also better delineate between the findings from the group’s previous work and the findings form the current study. Finally, authors should be cautious about describing data as “perfectly comparable” to previous studies.

“Finally” is spelled incorrectly in the discussion.

Reviewers' comments:

Reviewer's Responses to Questions

**Comments to the Author**

1. Is the manuscript technically sound, and do the data support the conclusions?

Reviewer #1: Partly

Reviewer #2: Yes

2. Has the statistical analysis been performed appropriately and rigorously? 

Reviewer #1: Yes

Reviewer #2: Yes

3. Have the authors made all data underlying the findings in their manuscript fully available?

Reviewer #1: Yes

Reviewer #2: Yes

4. Is the manuscript presented in an intelligible fashion and written in standard English?

Reviewer #1: No

Reviewer #2: Yes

5. Review Comments to the Author

Reviewer #1: The rationale and objective of this study is clear. Moreover, authors successfully describe the contribution of this work to the field, while recognizing its limitations and noting outstanding questions. However, the manuscript requires much improvement to the writing quality and data presentation and interpretation. The manuscript lacks organization and focus due to run-on and awkwardly structured sentences. In addition, figures are missing crucial details required to interpret the data. Several results are stated without a clear connection to the supporting data.

Please see attached document for remaining comments.

Reviewer #2: Bertrand and colleagues conducted a retrospective, multicenter, study to test potential correlation between the C3D-fixing antibody and IgG SAB by Imucor in a cohort of patients previously tested by the IgG SAB assay by One Lambda (MFI > 1000).

The authors confirmed previous report showing some differences between data obtained by the two vendors (Immucor and One Lambda), looking either at immunedominant DSA or sum of DSAs. They further attempted comparison between histological diagnosis on the biopsy and antibody characteristics and show no correlation

Biopsies were obtained with significant time difference from sera test (within 6 months for some samples) and the purpose for obtaining them is not clear – does not look like the same time point from transplant for all patients – the only thing in common is that they did not correlate with graft dysfunction.

Overall, this is a relatively small study cohort, utilizing patients from 9 different centers, with no systematic approach to obtaining either biopsies or serum samples. With that – it is not clear what the clinical value of their finding is.

6. PLOS authors have the option to publish the peer review history of their article (what does this mean?). If published, this will include your full peer review and any attached files.

Reviewer #1: No

Reviewer #2: No

---

## [Author Response · Author response to Decision Letter 0]

16 Mar 2021

The reviewer # 1 requested that the authors make more conclusive statement based on their analysis. Indeed, the statement that more study should be performed is not satisfactory as almost every study needs a continuation. 

Thank you for your analysis. We tried to make more conclusive statement in our revised manuscript. 

The Reviewer # 2 indicated at several editorial changes which need to be addressed. 

• Readers could benefit from a more detailed description of HLA donor specific antibodies

We added in the introduction a more detailed description on HLA DSA and dn DSA

• and de novo DSA in the introduction, and how this is relevant to transplant pathology.

We added in the introduction a more detailed description on HLA DSA and dn DSA.

• Please explain rationale for accepting positive results for the IM Lifecodes LSAB after 2 of 3 criteria were met, instead of all requiring all criteria to be met.

Thank you for your comment. Accepting positive results for the IM Lifecodes LSAB after 2 of 3 criteria were met is used according to the manufacturers' instructions. Nevertheless, when BCM is low (below 1500) but the amount of antigen in the bead is also low, result could be positive because BCR and AD-BCR are positive.

• Please indicate negative control beads for the IM Lifecodes LSAB.

As indicated in the methods section, the negative control beads typically yield low MFI values (lot specific) when treated with the positive or negative control sera.

• Authors should specify concentration of beads, phycoerythrin conjugated goat antihuman IgG, anti-human C3d, instead of, or in addition to the volume used, as volume is arbitrary without knowing this information.

We completely understand your comment. Unfortunately, all these components are provided with LIFECODES LSAB and LIFECODES C3d kits by the manufacturer and the concentration of beads, phycoerythrin conjugated goat antihuman IgG, anti-human C3d are not available. 

• Please describe methods for light microscopy and immunofluorescence staining of kidney tissue.

Thank you for your comment. Light microscopy analysis was performed after fixation in Dubosq Brazil solution, 2-μm paraffin section, and coloration by Masson trichrome, hematoxylin & eosin, periodic acid-Schiff and Marinozzi silver stainings. C4d staining was performed on paraffin sections by immunohistochemical (IHC) analysis using a rabbit anti-human C4d polyclonal IgG, or by immunofluorescent staining performed on 4-μm frozen sections with monoclonal mouse anti-human C4d antibody. We added this point in the revised manuscript.

• In statistical analysis, authors should indicate a Student t-test was used.

This point was added in the statistical analysis section.

• Please describe the advantages/disadvantages of the iDSA and DSA parameters. For example, why were both used, and is one superior to the other?

Thank you for your comment. Both iDSA and sDSA were used in this study because some kidney transplant recipients had more than 1 DSA. Even if the intensity of the iDSA is not sufficient to predict subclinical ABMR, the combination of all DSA would be. 

Réf: Bertrand D, Farce F, Laurent C, et al. Comparison of Two Luminex Single-antigen Bead Flow Cytometry Assays for Detection of Donor-specific Antibodies after Renal Transplantation. Transplantation. 2019;103(3):597–603. doi:10.1097/TP.0000000000002351

• Please indicate all abbreviations used in Table 1.

We indicate all abbreviations used in Table 1 in the revised manuscript.

• Authors should consider reformatting the methods and results sections from bullet points to paragraphs.

Thank you for your help to improve the methods and results sections, presented in the revised manuscript.

• For reader’s ease, authors should consider labeling Figure 1 panels A-C to indicate the correlation described (as indicated in the Figure legend).

We corrected this point in the manuscript

• For Figure 2, please define the descriptive statistics reflected in the box and whisker plot.

We defined the descriptive statistics reflected in the box and whisker plot.

• Please label y-axes in Figures 2, 3, 4, 5, and 6.

We labeled y-axes in Figures 2, 3, 4, 5, and 6.

• Please specificy what the numbers refer to on the top of each bar in Figure 3. In addition, please explain in more detail the significance of Figure 3.

Thank you for your comment. We made these corrections in the revised manuscript.

• Figure 4 is not clearly described and thus, difficult to interpret. To address this, please indicate the abbreviations used (g+cpt, i+t, ci+ct, cv, and cg) in the figure legend. Please define the numbers indicated on top of the bar graphs in the figure legend.

Thank you for your comment. We made these corrections in the revised manuscript.

• Authors should indicate the Figure showing the data supporting the following results: “The proportions of patients with positive C3d-fixing DSA were not statistically different in patients with active sABMR, chronic active sABMR or without sABMR (56.7% vs 47.1 % vs 57.1%, p=NS).”

Thank you for your comment. We did not add an extra figure for this point meaning that it was not useful for the readers.

• Please indicate the figure showing the data supporting the following results that begin with the sentence: “The intensity (BCM) of iDSA and sDSA were statistically higher in the C3dfixing DSA group…”

Thank you for your comment. Figure 7 about the intensity (BCM) of iDSA and sDSA according C3d fixing DSA positivity was added in the revised manuscript.

• Please be consistent with reporting non-significant results as either “NS” (without a p-value) or reporting a p-value greater than 0.05. Preference is to report p-values. Authors should also consider reporting actual p-values for significant data (instead of indicating p<0.0001).

We modified this point in the new manuscript.

• This sentence is awkwardly written: “In our previous published study 4, we demonstrated that this strategy led to the diagnosis of sABMR in up to 40 %.”

We completely agree with your comment. We modified this sentence in the text. 

• The discussion should be condensed to be more concise and direct because it lacks clear focus. The discussion should also better delineate between the findings from the group’s previous work and the findings form the current study. Finally, authors should be cautious about describing data as “perfectly comparable” to previous studies.

Thank you for this comment. We modified the discussion in order to be more concise and direct. We also tried to delineate between the findings from our group’s previous work and the findings form the current study.

• “Finally” is spelled incorrectly in the discussion.

We modified this point in the new manuscript.

---

## [Editor Report · Decision Letter 1]

29 Mar 2021

Intensity of de novo DSA detected by Immucor Lifecodes Assay and C3d fixing antibodies are not predictive of subclinical ABMR after Kidney Transplantation.

PONE-D-21-00625R1

Dear Dr. Bertrand,

We’re pleased to inform you that your manuscript has been judged scientifically suitable for publication and will be formally accepted for publication once it meets all outstanding technical requirements.

Kind regards,

Stanislaw Stepkowski

Academic Editor

PLOS ONE
---

## [Editor Report · Acceptance letter]

12 Apr 2021

PONE-D-21-00625R1 

Intensity of de novo DSA detected by Immucor Lifecodes Assay and C3d fixing antibodies are not predictive of subclinical ABMR after Kidney Transplantation. 

Dear Dr. Bertrand:

I'm pleased to inform you that your manuscript has been deemed suitable for publication in PLOS ONE. Congratulations! Your manuscript is now with our production department. 

Kind regards, 

on behalf of

Dr. Stanislaw Stepkowski 

Academic Editor

PLOS ONE